# Delta-Radiomics Based on Dynamic Contrast-Enhanced MRI Predicts Pathologic Complete Response in Breast Cancer Patients Treated with Neoadjuvant Chemotherapy

**DOI:** 10.3390/cancers14143515

**Published:** 2022-07-20

**Authors:** Liangcun Guo, Siyao Du, Si Gao, Ruimeng Zhao, Guoliang Huang, Feng Jin, Yuee Teng, Lina Zhang

**Affiliations:** 1Department of Radiology, The First Affiliated Hospital of China Medical University, Shenyang 110001, China; guolc@cmu.edu.cn (L.G.); sydu@edu.cmu.cn (S.D.); gaosi@cmu.edu.cn (S.G.); rmzhao@cmu.edu.cn (R.Z.); glhuang@cmu.edu.cn (G.H.); 2Department of Breast Surgery, The First Affiliated Hospital of China Medical University, Shenyang 110001, China; jinfeng@cmu.edu.cn; 3Departments of Medical Oncology and Thoracic Surgery, The First Affiliated Hospital of China Medical University, Shenyang 110001, China

**Keywords:** DCE-MRI, breast cancer, radiomics, neoadjuvant chemotherapy, pathological complete response

## Abstract

**Simple Summary:**

Neoadjuvant chemotherapy (NAC) followed with surgery is the standard strategy in the treatment of locally advanced breast cancer, but the individual efficacy varies. Early and accurate prediction of complete responders determines the NAC regimens and prognosis. Breast MRI has been recommended to monitor NAC response before, during, and after treatment. Radiomics has been heralded as a breakthrough in medicine and regarded to have changed the landscape of biomedical research in oncology. Delta-radiomics characterizing the change in feature values by applying radiomics to multiple time points, is a promising strategy for predicting response after NAC. In our study, the delta-radiomics model built with the change of radiomic features before and after one cycle NAC could effectively predict pathological complete response (pCR) in breast cancer. The model provides strong support for clinical decision-making at the earliest stage and helps patients benefit the most from NAC.

**Abstract:**

Objective: To investigate the value of delta-radiomics after the first cycle of neoadjuvant chemotherapy (NAC) using dynamic contrast-enhanced (DCE) MRI for early prediction of pathological complete response (pCR) in patients with breast cancer. Methods: From September 2018 to May 2021, a total of 140 consecutive patients (training, *n* = 98: validation, *n* = 42), newly diagnosed with breast cancer who received NAC before surgery, were prospectively enrolled. All patients underwent DCE-MRI at pre-NAC (pre-) and after the first cycle (1st-) of NAC. Radiomic features were extracted from the postcontrast early, peak, and delay phases. Delta-radiomics features were computed in each contrast phases. Least absolute shrinkage and selection operator (LASSO) and a logistic regression model were used to select features and build models. The model performance was assessed by receiver operating characteristic (ROC) analysis and compared by DeLong test. Results: The delta-radiomics model based on the early phases of DCE-MRI showed a highest AUC (0.917/0.842 for training/validation cohort) compared with that using the peak and delay phases images. The delta-radiomics model outperformed the pre-radiomics model (AUC = 0.759/0.617, *p* = 0.011/0.047 for training/validation cohort) in early phase. Based on the optimal model, longitudinal fusion radiomic models achieved an AUC of 0.871/0.869 in training/validation cohort. Clinical-radiomics model generated good calibration and discrimination capacity with AUC 0.934 (95%CI: 0.882, 0.986)/0.864 (95%CI: 0.746, 0.982) for training and validation cohort. Delta-radiomics based on early contrast phases of DCE-MRI combined clinicopathology information could predict pCR after one cycle of NAC in patients with breast cancer.

## 1. Introduction

As a standard care for locally advanced breast cancer, neoadjuvant chemotherapy (NAC) is clinically useful to downsize and downstage tumors and reduce the extent of surgery from mastectomy to breast conservation. Less than 10% to 50% of patients achieve pathological complete response (pCR) through NAC depending on receptor status and subtype [1]. Achieving pCR can result in a more favorable long-term prognosis [2,3,4]. If we can confidently predict that a patient has a high probability of pCR, surgery can be safely postponed or even omitted. Thus, predicting the likelihood of pCR is important for the development of improved and personalized treatment plans. The earlier accurate predictions are made, the more likely patients are to benefit. Therefore, early and reliable predictors of tumor response are needed.

Breast MRI is recommended to monitor therapy response during NAC [5,6], with dynamic contrast-enhanced MRI (DCE-MRI) providing vascular information to evaluate tumor presence with high sensitivity. Tumor morphology, vascularity, and cell density can be affected by NAC. Many feature-level MRI metrics have been explored to monitor treatment response, including tumor size [7,8,9,10], DCE kinetic parameters [7,11,12], diffusion measures [8,13], and MRI texture parameters [14,15,16]. Recent studies [17,18,19,20] have demonstrated the feasibility of radiomics for predicting NAC response. Most radiomics studies about breast cancer [18,19,20] extracted radiomic features from single time-point images (e.g., before NAC). However, these studies generally neglect the changes in tumors during NAC. Delta-radiomics is a new concept based on the changes in radiomic features in a set of longitudinal images [21]. Therapy-induced changes in tumor morphology and heterogeneity can be quantified using delta-radiomics as a complement to response evaluation criteria in solid tumor (RECIST) for monitoring therapeutic response in several tumor types [22,23]. In the field of breast cancer, key radiomic features associated with the pathological response were significantly different before and after NAC [24]. Fan et al. [25] evaluated the delta-radiomic model after the second cycle of NAC using DCE-MRI. However, whether the delta-radiomics model can predict NAC response after the first cycle, which is the earliest time-point, remains unclear.

We assumed that delta-radiomics could reflect changes in tumor heterogeneity or genetic profiles after only one cycle of NAC. The main objective of this work was to determine whether therapy-induced delta-radiomics features can improve models for predicting NAC outcomes when used in conjunction with longitudinal fusion radiomic features. Additionally, we also used differential subsampling with cartesian ordering (DISCO) DCE-MRI, which has high spatiotemporal resolution [26], to analyze the model performance in the early, peak, and delay phases during contrast agent inflow and outflow, and determine the optimal contrast phase of DCE-MRI.

## 2. Materials and Methods

### 2.1. Patients

The prospective protocol was approved by our Institutional Review Board (approval code: 2019-33-2), and each participant provided written informed consent.

In the prospective study, the inclusion criteria were as follows: (1) the patient had biopsy-proven unilateral primary breast invasive ductal cancer and ipsilateral axillary lymph node metastasis without prior treatment; (2) complete biopsy information including histological grades, receptor status, and Ki-67 level of the primary tumor was available; and (3) baseline DCE-MRI and DCE-MRI following the first cycle of NAC were conducted. The exclusion criteria were as follows: (1) occult cancers or small lesions less than 1.0 cm in diameter on baseline DCE-MRI; (2) large lesions more than 10.0 cm in diameter; (3) incomplete NAC cycle or change in chemotherapy regimen during NAC; (4) surgery after NAC was performed in an external institution. Figure 1 shows a flowchart of patient collection. From March 2019 to May 2021, the consecutive cohort enrolled in our study included a total of 162 patients with paired DCE-MRI (baseline DCE-MRI and DCE-MRI following the first cycle of NAC). According to the exclusion criteria, 22 patients were excluded due to occult primary foci (*n* = 2), small lesions (*n* = 6), oversized lesions (*n* = 3), incomplete NAC cycle (*n* = 3), change in chemotherapy regimen (*n* = 6), and undergoing surgery at an external institution (*n* = 2). Based on their admission time, the remaining 140 patients were split into a training cohort and a validation cohort at a ratio of 7:3 (98 patients for training and 42 patients for validation).

### 2.2. DCE-MRI Data Acquisition

Pretreatment MRI was performed within one week before NAC. Follow-up MRI after the first cycle of NAC was performed within 72 h before the second cycle of NAC. Both breast MRI examinations were performed using a 3 T MR scanner (SIGNATM Pioneer, GE Healthcare, Milwaukee, WI, USA) with an 8-channel phased-array breast coil. T1-weighted DCE-MRI sequence (one pre-contrast phase and 20 post-contrast phases with temporal resolution of 19.4 s) was obtained using three-dimensional (3D) DISCO and fat suppression technique (GE Healthcare). The scanning parameters were as follows: TR = 4.9 ms, TE = 1.7 ms, flip angle = 90°, FOV = 360 × 360 mm, matrix = 256 × 256, section thickness = 1.4 mm, intersection gap = 1.4 mm, number of sections = 120/phase, acceleration factors = 2. After the first pre-contrast scanning followed by a pause of 20 s, the contrast agent was injected intravenously as a bolus (0.1 mmol/kg body weight) by a power injector at 2 mL/s followed by a 20 mL saline flush. Subsequently, 20 phase post-contrast images were acquired.

### 2.3. Clinicopathologic Features

Estrogen receptor (ER), progesterone receptor (PR) and human epidermal growth factor receptor 2 (HER2) were evaluated according to ASCO/USCAP guidelines [27,28] using an avidin–biotin immunohistochemistry technique for biopsy specimens before NAC. The Ki-67 index was assessed with a cut-off value of 20% [29]. The molecular subtype was categorized according to the 2017 St. Gallen guidelines [30]. Patients with hormone receptor (HR)-positive, low Ki-67, and HER2-negative in breast tumors were defined as luminal A subtype. Luminal B subtype included patients with HR-positive, high Ki-67, or HER2-positive. If HR-negative, HER2-enriched subtype patients were distinguished by HER2 overexpression or overamplification, whereas those with HER2-negative breast tumors were classified as triple-negative subtype patients. For multi-lesion patients, the receptor state of the largest lesion was selected for assessment.

### 2.4. NAC Regimen and Criteria for pCR

According to the National Comprehensive Cancer Network guideline [31], all participants received the standard six or eight cycles of NAC before surgery. The NAC regimens were based on taxane, anthracycline, or both anthracycline and taxane. For HER2-positive tumors, anti-HER2-targeted trastuzumab or trastuzumab + pertuzumab were added to the chemotherapy drugs (8 mg/kg as the loading dose and 6 mg/kg as the maintenance dose).

Treatment response was evaluated after all NAC cycles based on surgical specimens. pCR was defined as the absence of residual invasive tumor (Miller–Payne grade 5, residual ductal carcinoma in situ could be present) and the absence of lymph node invasion in the ipsilateral sentinel node or lymph nodes removed during axillary dissection (ypT0/isN0) [3,8,13,14,18,32].

### 2.5. Tumor Segmentation and Repeatability Analysis

The radiomics workflow is illustrated in Figure 2. For the pre-treatment and post-treatment DCE-MRI for all patients, the open-source ITK-snap software (www.itksnap.org, version 3.8.0) (accessed on 28 July 2020) was imported to segment the volume of interest (VOI) [33]. Each tumor lesion was semi-automatically segmented on the peak contrast phase (8th post-contrast phase according to time intensity curve [TIC]) of DCE-MRI (CEp). By using region growing methods, the tumor VOIs covered the tumor areas. If deemed necessary, manual adjustments were made dominantly in the axial position, auxiliary by the coronal and sagittal positions. Necrosis and blood areas were included in the tumor VOIs. If the tumor was a unilateral multifocal and multicentric lesion, the largest one was selected as the object. The tumor VOIs on the CEp images were propagated with slight adjustment to the early contrast phase of DCE-MRI (CEe; 5th post-contrast phase according to TIC) and the delay contrast phase of DCE-MRI (CEd; 18th post-contrast phase according to TIC) (Appendix A).

Two radiologists with two and five years of experience in breast cancer diagnosis independently performed VOI delineation to test intra-observer reproducibility. They independently segmented the pretreatment MRI CEp images of breast cancer in 30 randomly selected samples. The radiomic features extracted from the above two VOIs were compared using the intra-class correlation coefficient (ICC). An ICC value of 0.8 or greater was considered to indicate almost perfect consistency, and the feature was retained. Features with ICC values less than 0.8 were initially eliminated. Then, the VOIs delineated by the radiologist with two years of experience were used as the final segmentation.

### 2.6. Radiomic Features and Their Changes

The radiomic features of the DCE-MRI images before NAC and after the first cycle were extracted using Analysis Kit software (A.K., GE Healthcare). For each VOI, the extracted features included seven categories of original image features: (1) first-order features (*n* = 18); (2) 2D and 3D shape features (*n* = 14); (3) gray-level co-occurrence matrix features (GLCM; *n* = 24); (4) gray-level run length matrix features (GLRLM, *n* = 16); (5) gray-level size zone matrix features (GLZSM, *n* = 16); (6) neighboring gray-tone difference matrix features (NGTDM, *n* = 5); (7) gray-level dependence matrix features (GLDM, *n* = 14); and (8) their wavelet-transformed type (*n* = 744). There is a total of 851 radiomic features, which have been used in previous studies [18,24,25,32,34]. Detailed names and definitions of all features can be found in Appendix A.

The changes in the radiomic features (delta-radiomics features) between CEe, CEp, and CEd were calculated from the differences between the pre-NAC features values (pre-radiomics features) and the 1st-NAC features values (1st-radiomics features):Delta-radiomics features = (pre-radiomics features) − (1st-radiomics features)(1)

### 2.7. Feature Selection

All patients in the training cohort were used to select features and build the prediction model. Separate pre-, 1st-, and delta-radiomics features under different contrast phases were selected. Before dimensionality reduction, features with variance ≤ 1 were excluded from analyses. The data were standardized (Z-score) using the following equation:Standardized value = (original value − average value)/standard deviation(2)

To obtain the features that were most strongly associated with pCR in the training cohort, we first performed Student’s *t*-test and univariate logistic regression analysis, and features with *p* < 0.1 were used for subsequent analysis. Spearman correlation analysis was then used to remove the features highly correlated with other features (the Spearman |*p*| ≥ 0.9). Finally, the least absolute shrinkage and selection operator (LASSO), was used for fine feature selection. A 5-fold cross-validation was used to tune the parameters to find the best λ value. Since LASSO is a regularization method, it can reduce the regression coefficients of features that are considered as attribute noise to zero and identify features that are non-redundant and robust. Thus, LASSO can overcome overfitting and improve the generalization capability of the proposed machine learning model [35].

### 2.8. Establishment and Performance of Models

#### 2.8.1. Separate Radiomic Models

In the training cohort, nine separate models under pre-, 1st-, delta-radiomics and different contrast phases were trained using multivariate logistic regression and 5-fold cross-validation, including model 1: pre-radiomics based on CEe; model 2: pre-radiomics based on CEp; model 3: pre-radiomics based on CEd; model 4: 1st-radiomics based on CEe; model 5: 1st-radiomics based on CEp; model 6: 1st-radiomics based on CEd; model 7: delta-radiomics based on CEe; model 8: delta-radiomics based on CEp; model 9: delta-radiomics based on CEd. The 5-fold cross-validation was used to optimize the tuning parameters to construct the multivariate logistic regression model. The validation cohort was used to evaluate the model performance. The radiomic score (Rad-score) for each patient in the training and validation cohort was computed. To assess the prediction performance, the receiver operating characteristic (ROC) curves were constructed, and the area under the ROC curves (AUCs) were calculated using the Rad-score. The performances of the separate pre-, 1st-, and delta-radiomics models under different contrast phases were compared using DeLong tests. The model with the largest AUC values and the AUC value more than 0.80 in the validation cohort was identified as the optimal model. The contrast phase of the optimal model was considered the optimal contrast phase for pCR classification.

#### 2.8.2. Longitudinal Fusion Radiomic Models

Longitudinal fusion radiomic models were established by combining longitudinal radiomic features based on the optimal contrast phases with the purpose of enhancing the predictive performance. After feature fusion, the methods of feature selection, model establishment, and model validation were the same as those for the separate models.

#### 2.8.3. Clinical-Radiomics Models

The clinical-radiomic model was established based on the Rad-scores of the separate and longitudinal fusion radiomic models under optimal contrast phase along with significant clinical indicators to explore any improvement resulting from feature fusion. The models were trained, and validated using the same strategy used to develop the separate radiomic models as described above. The AUC, sensitivity, specificity, and accuracy of clinical-radiomics models will be calculated and presented.

### 2.9. Statistical Analysis

Clinicopathological characteristics and pretreatment MRI findings were compared between pCR and non-pCR using independent *t*-test or Mann–Whitney U test for continuous variables and chi-square test or Fisher’s exact test for categorical variables. The ROC curve was constructed to determine model performance based on the AUC, accuracy, sensitivity, and specificity. Calibration curves were used for each model to depict the agreement between the predicted probability of pCR and the observed outcomes. The Hosmer–Lemeshow test was used to determine the goodness of fit of the models, and *p* values of more than 0.05 were considered well calibrated. Decision curve analysis (DCA) was used to evaluate clinical utility by quantifying the net benefits of the training and validation cohort. The DeLong test was applied to compare the differences in the AUC values of different models. Heatmaps were generated to show the distribution between pCR and non-pCR. All statistical analyses were performed with R 4.1.1 and Python 3.70 (https://www.python.org/) (accessed on 28 July 2020). The R packages used in this study include “caret,” “glmnet,” “pROC,” “InformationValue,” “leaps,” and “bestglm.” A two-tailed *p*-value < 0.05 indicated statistical significance.

## 3. Results

### 3.1. Clinical Characteristics

A total of 140 lesions from 140 women (mean age, 50.51 years; age range, 28–73 years) were ultimately evaluated. Out of the 98 patients in the training cohort, 28 (28.6%) achieved pCR, while 12/42 patients (28.6%) in the validation cohort achieved pCR. For the classification of pCR, estrogen receptors, HER2 status, and molecular subtypes in both training and validation cohort, and progesterone receptors in training cohort were significantly different. The differences in the other clinical features and MRI morphology parameters were not statistically significant (Table 1).

### 3.2. Repeatability Analysis

The ICCs for all radiomic features and their changes were greater than 0.80 between the two radiologists.

### 3.3. Separate Radiomic Models

Table 2 shows the performances of the separate radiomic models under different contrast phases.

The detailed features selected are described in Appendix A. For all models, the optimal performance appeared in the delta-radiomics model, which gave AUC values of 0.917 (95% CI: 0.861, 0.974) for the training cohort and 0.842 (95% CI: 0.709, 0.974) for the validation cohort using nine selected features under CEe. The delta-radiomics model performed better than the pre-radiomics model, which achieved AUC values of 0.759 (95% CI: 0.647, 0.871) for the training cohort and 0.617 (95% CI: 0.403, 0.830) for the validation cohort (DeLong test: *p* = 0.011/0.047 for training/validation cohort). The performance of 1st-radiomics models under different contrast phases was moderate; the optimal AUC values were 0.803 (95% CI: 0.694, 0.913) for the training cohort and 0.775 (95% CI: 0.627, 0.923) for the validation cohort under CEe. The delta-radiomics model under CEe performed better than the optimal 1st-radiomics model under CEe (DeLong test: *p* = 0.054/0.500 for the training/validation cohort). Figure 3 shows the LASSO selection process, and Figure 4a presents the results of logistic regression for the delta-radiomics model.

### 3.4. Determination of the Optimal Contrast Phase

By comparing the model performance, we found that the delta-radiomics model based on the changes in radiomic features achieved superior performance compared to the other models, and CEe was the optimal contrast phase for pCR classification. The ROC curves of all the separate radiomic models are shown in Appendix A.

### 3.5. Longitudinal Fusion Radiomic Model under the Optimal Contrast Phase

Under the optimal contrast phase (CEe), the delta-radiomics features were separately fused with the pre- and 1st-radiomics features to build longitudinal fusion models. Table 3 shows the performances of the longitudinal fusion models. For the fusion of delta- and pre-radiomics features, the selected five features were three delta-radiomics features and two pre-radiomics features, which achieved AUC values of 0.866/0.750 for the training/validation cohort. The corresponding OR values and coefficients of key features for the fusion radiomic models are shown in Figure 4b.

When the delta- and 1st-radiomics features were fused, six of the selected seven features were delta-radiomics features (Figure 4c). The resulting fusion model achieved AUC values of 0.903/0.839 for the training/validation cohort. This fusion model did not show improved performance compared to the delta-radiomics model (AUC = 0.917/0.842 for the training/validation cohort).

A total of five features were conserved from the fusion model of the pre-, 1st-, and delta-radiomics features. Compared with the delta-radiomics model (AUC = 0.917/0.842 for the training/validation cohort), the AUC values (AUC = 0.871/0.869 for the training/validation cohort) showed a slight improvement for the validation cohort. Among the selected features, three were delta-radiomics features, one was a 1st-radiomics feature, and one was a pre-radiomics feature. The corresponding OR values and coefficients are shown in Figure 4d. The longitudinal fusion model showed good agreement between the actual observations and classifications in both the training cohort (Figure 5a) and validation cohort (Figure 5b). Nonsignificant statistics were achieved in the Hosmer–Lemeshow test in the training cohort (*p* = 0.388) and validation cohort (*p* = 0.185). DCA showed that the radiomic model would add more benefit in distinguishing pCR and non-pCR when the threshold probability was at any threshold in the training cohort (Figure 5c) or between 15% to 82% in the validation cohort (Figure 5d).

### 3.6. Clinical-Radiomics Models

After adding significant clinical features (HR and HER2) to the radiomic models, the delta-radiomics model still produced the highest AUC (AUC = 0.934/0.864, sensitivity = 0.927/0.750, specificity = 0.829/0.867, accuracy = 0.857/0.833 for the training/validation cohort). Figure 6 shows the ROC curves for the optimal separate radiomic models, longitudinal fusion radiomic models, and clinical-radiomics model. The calibration curves and DCA results of the training set for the clinical-radiomics models are shown in Figure 5. The clinical-radiomics model showed good diagnostic performance based on the actual observations and classifications in the training cohort (Figure 5a) and validation cohort (Figure 5b). Nonsignificant statistics were achieved in the Hosmer–Lemeshow test in the training cohort (*p* = 0.910) and validation cohort (*p* = 0.410). DCA indicated that the clinical-radiomics model was most beneficial in distinguishing pCR and non-pCR when the threshold probability was between 0 and 0.82 in the training cohort (Figure 5c) or at any given threshold probability in the validation cohort (Figure 5d).

### 3.7. Changes in Radiomic Features in the Delta-Radiomics Model

The nine selected radiomic features in the delta-radiomics model and their values before NAC and after the first cycle of NAC are shown in Table 4. The nine radiomic features included two describing the heterogeneity of the high-gray region, one describing the heterogeneity of the low-gray region, two describing the local homogeneity of the image, one measuring the skewness and asymmetry of GLCM, one measuring the average gray-level intensity within the VOI, one reflecting the similarity between gray-level intensity values, and one gray value–voxel correlation parameter. The feature reflecting the similarity between the gray intensity values (wavelet—LLH_ glrlm_ Gray Level Non-Uniformity Normalized) was upregulated after early treatment, and have higher levels in the pCR group. All other characteristics were downregulated after early treatment, with a greater decrease in the pCR group. Eight of the nine features were wavelet-transformed types. Four features (wavelet—HHH_ glszm_ Large Area Low Gray Level Emphasis, wavelet—HLH_ gldm_ Large Dependence High Gray Level Emphasis, and wavelet—LHH_ gldm_ Large Dependence High Gray Level Emphasis and wavelet—LLH_ first order_ Mean) from wavelet-transformed type describing the information of gray level. An example of a wavelet-transformed feature wavelet—HLH_ gldm_ Large Dependence High Gray Level Emphasis (OR = 2.164), which is also retained in the longitudinal fusion radiomic models and is an important feature in the fusion model of pre- and delta-radiomics features (OR = 3.213), the fusion radiomic model of 1st- and delta-radiomics features (OR = 3.213), and the fusion radiomic models of pre-, 1st-, and delta-radiomics features (OR = 3.936). Figure 7 shows the changes in this feature between pre-NAC and 1st-NAC.

## 4. Discussion

The early prediction of treatment response of locally advanced breast cancer to NAC is important for optimizing and adjusting the treatment plan. Our study demonstrated that DCE-MRI-based delta-radiomics after the first cycle of NAC can quantify changes in tumor heterogeneity at the earliest time point. It improved prediction ability using single time-point images (e.g., pre-NAC). Additionally, by comparing the early, peak, and delayed DCE-MRI radiomic models, the optimal contrast phase was preliminarily confirmed to be the early contrast phase, providing a reference for extracting reliable and comparable DCE-MRI data for subsequent radiomics studies.

Many studies [18,19,20] have focused on pre-NAC radiomics for pCR prediction in breast cancer. In studies on multimodal radiomics, pre-radiomics models based on contrast-enhanced T1-weighted images did not show surprising performance for pCR classification [18,19,20]. In a large multi-center study [18], Liu et al. found that the AUC achieved by DCE-MRI in the peak phase was less than 0.60. DCE-MRI remains the most essential among a wide range of technologies because its high spatial resolution allows it to precisely reveal tumor size, shape, and internal heterogeneity, thus facilitating the tumor segmentation. Breast DCE-MRI for evaluating the treatment response is recommended every two NAC cycles by expert consensus [36]. The changes in tumor size [8,9,37] and functional parameters [7,11] based on DCE-MRI after two NAC cycles and even after one cycle strongly predicted the final therapeutic response. Recent studies extended the longitudinal changes to texture [14,15,16] and radiomic features [25,26]. Delta-radiomics features after early treatment reflect therapy-induced changes in tumor morphology and heterogeneity, which may improve the performance of single pre-NAC images. Eun et al. [14] found that DCE-MRI texture-based models from a single time-point (after three or four cycles of NAC) had the highest diagnostic performance (AUC, 0.82; 95% CI: 0.74, 0.8). However, the middle stage of the NAC treatment is too late to make changes to the regimen in patients for whom NAC is ineffective. In another small-sample study [16], Nadrljanski et al. found that DCE-MRI performed after two NAC cycles could reveal the differences in texture features between patients that respond and do not respond to NAC. Advancing scanning time point after the first cycle of NAC, we extracted comprehensive radiomic features from DCE-MRI before and after the first cycle, analyzed their differences, and built an optimal model for the earlier and more accurate prediction of patient outcomes. Recent studies [38,39], which used deep learning and transfer learning for feature extraction and selection without human intervention, have been already successfully applied on pre-treatment and early-treatment DCE-MRI and achieved a good performance. Some difficulties have been recently overcome thanks to deep learning, such as time-consuming manual labeling, inconsistent DCE-MRI protocols, etc. Furthermore, fully automatic segmentation is not restricted to intratumoral features, breast tissue [40] and peritumoral [41] can also give an early prediction. This is the direction of our further efforts.

Although the DCE-MRI scanning sequence varied among studies, the early phase [14,19,20,37] and peak phase [18,19,25] are the most commonly used phases for extracting DCE-MRI radiomic features. Based on DCE-MRI under the peak contrast phase, Fan et al. [25] explored the delta-radiomics features after the second NAC cycle. The diagnostic power obtained using the first cycle data in our study (AUC = 0.764 for delta-radiomics) was similar to that achieved by the second cycle data of Fan et al. [25] (AUC = 0.726 for delta-radiomics). We extracted features under multiple contrast phases and found that the early phase performed better than the peak or delay phases. The reason may be that our DCE-MRI used the DISCO protocol. DISCO provides higher temporal and comparable spatial resolution compared with clinical standard protocol [26,42]. This advantage is beneficial for detection and classification of breast lesions with high accuracy [43,44]. High temporal resolution makes the phase capture more accurate, especially for malignant tumors with rapid early enhancement. High spatial resolution guarantees the accuracy of radiomic features because high spatial resolution leads to better classification compared with low spatial resolution [34]. Despite the noted strengths, the clinical application of DISCO is still in its infancy and remains exploratory. The first issue to address is the storage and transmission difficulties caused by big data volumes. The absence of clear guidelines for quantitative measurement is another current issue. Tumor segmentation in the different contrast phases influences DCE-MRI parameter measurement, and the early phase is optimal for the extraction of reliable DCE-MRI kinetic parameters [45]. The early phase of DCE-MRI is recommended for response monitoring following chemotherapy and pCR prediction [46]. Our findings further confirm that the early phase should be used in radiomics studies. The most informative feature values for tumor characterization should be available at two minutes or less after the injection of contrast agent [47]. A possible explanation for this might be that the cumulative difference of the contrast agents for each voxel in early phase highlighted tumor heterogeneity, while these differences tend to balance in the peak/delay phases. Thus, peak- or delayed-phase imaging may not provide the most valuable feature information for predicting pCR.

Features from delta-radiomics account for the largest proportion of longitudinal fusion radiomic models. This is consistent with our hypothesis: the delta-radiomics features reflected the therapy-induced changes in tumor heterogeneity, allowing the prediction of pCR. After adding clinical factors (HR and HER2 status) to the delta-radiomics model, the clinical-radiomics model achieved an AUC of 0.934/0.864 in the training/validation cohort. Thus, combining clinicopathological information with delta-radiomics is effective in predicting the performance of pCR in breast cancer patients. Focusing on the meanings of radiomic features, Zhou et al. [32] found that the features of wavelet-transformed feature showed promise for predicting pCR in response to NAC for patients with breast cancer, which supports our results. Among the key wavelet-transformed features based on the delta-radiomics model in our study, the three most important features were wavelet—LLH_ glcm_ Idn, wavelet—LLH_ firstorder _ Mean, and wavelet—HLH_ gldm_ Large Dependence High Gray Level Emphasis. All of these features were downregulated after early treatment and were significantly higher in pCR patients than in non-pCR patients. This can be explained by the different changes in tumor heterogeneity observed in the two response groups. Among the three most important wavelet-transformed features, two reflect the gray level of the entire tumor. The feature wavelet—HLH_ gldm_ Large Dependence High Gray Level Emphasis was important in all longitudinal fusion radiomic models and showed good robustness for predicting pCR. This further indicates that heterogeneous changes of high proliferative activity area with high-gray in DCE-MRI better cast light upon pCR. The large dependence high gray-level emphasis of Jacobian maps (a registered map that reflects the level of voxelwise volumetric shrink or expansion) in Fan et al. [25] also found to be higher in non-responders than responders after the second cycle of NAC. In the current study, considering that Jacobian map did not show better performance than the delta-radiomics models, we analyzed feature-level delta-radiomics features without voxelwise-level changes. Jahani et al. [48] reported the changes in voxelwise first-order features resulted in better pCR prediction compared with feature-level changes. It is worth applying radiomics to maps of voxelwise features changes in future research. We also found that the feature original— glcm_ Correlation, which reflects the correlation of local gray levels in the image, was also important in the delta-radiomics model and the fusion models. The value of this feature was downregulated after early treatment and was significantly higher in pCR patients than in non-pCR patients. Thus, gray scale related features also play an important role in predicting pCR.

Despite the considerable diagnostic power of delta-radiomics using longitudinal images, several limitations of this pilot study should be acknowledged. First, the sample size of the prospective cohort was relatively small. Further studies with large numbers of patients from multiple centers should be conducted to confirm our findings. Second, the different cancer subtypes included in our study are treated by different NAC regimens, although we affirmed that all NAC regimens were standard. Third, we only detected feature-level changes; we did not evaluate voxel-level changes. Fourth, DISCO protocol with high spatiotemporal resolution is conducive to quantitative and semi-quantitative measurements, while we did not compare or combine radiomics with quantitative parameters. Finally, there is no clear biological explanation for radiomic features and their changes, and interpretability remains the biggest limitation of current radiomics studies.

## 5. Conclusions

Delta-radiomics characterizing the change in feature values by applying radiomics to multiple time points, is a promising strategy for predicting pCR after NAC. Using DCE-MRI evaluations before and after one NAC cycle, delta-radiomics under the early contrast phase combined with clinicopathological information has excellent predictive power for pCR prediction.

## Figures and Tables

**Figure 1 cancers-14-03515-f001:**
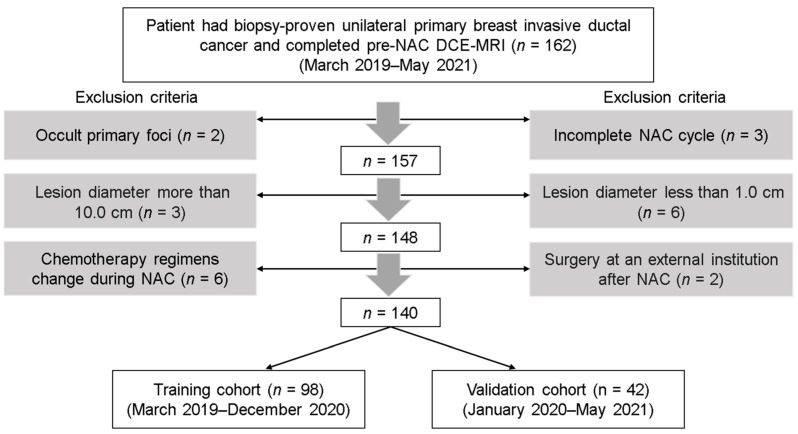
Flowchart of patient collection. NAC, neoadjuvant chemotherapy; DCE-MRI, dynamic contrast-enhanced.

**Figure 2 cancers-14-03515-f002:**
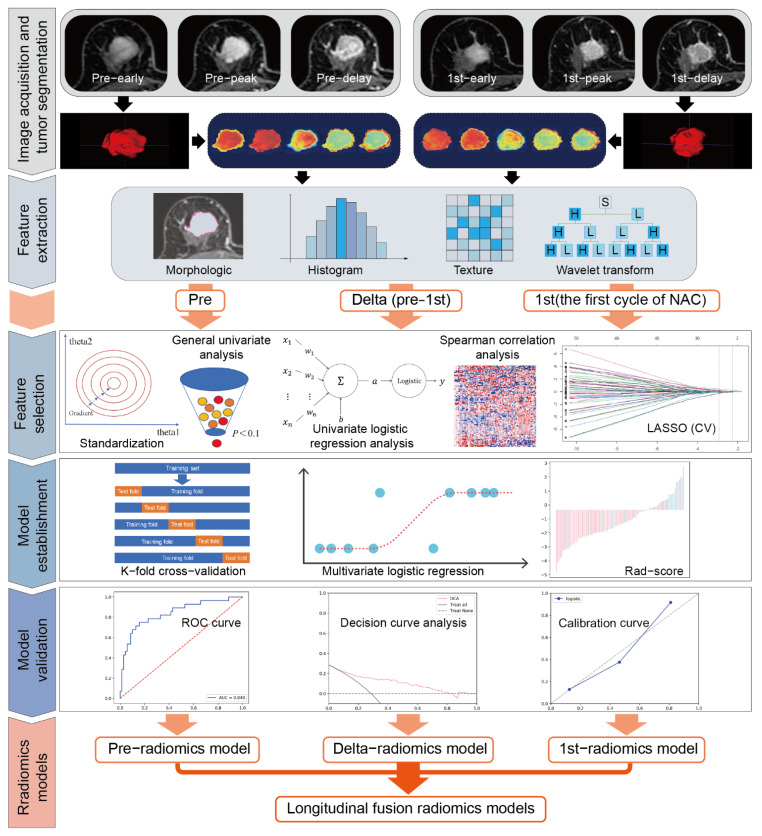
Workflow of the radiomics analysis. Pre, pre-NAC; 1st, the first cycle of NAC; CEe, the early phase of DCE-MRI; CEp, the peak phase of DCE-MRI; CEd, the delay phase of DCE-MRI; Delta, delta-radiomics features (the relative change values between pre- and 1st-radiomics features); LASSO, the least absolute shrinkage and selection operator; CV, cross-validation; ROC curve, the receiver operating characteristic curve.

**Figure 3 cancers-14-03515-f003:**
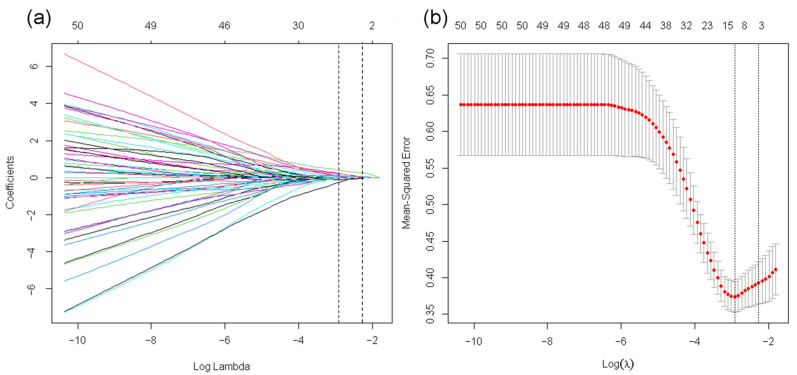
Feature selection for LASSO of the delta-radiomics features. (**a**) Tuning parameter (λ) selection by 5-fold cross-validation with minimum criteria. Binomial deviance (*y*-axis) was plotted against log(λ) (*x*-axis). The left dotted vertical line was at the optimal lambda value point by using the minimum criteria, and the right line was at the optimal lambda value point by using one standard error of the minimum criteria (the 1-SE criteria). The optimal value of λ was 0.054 (minimum), and the corresponding value of log(λ) = −2.919. (**b**) The least absolute shrinkage and selection operator coefficient profiles of the 80 radiomic features after univariate analysis. For the optimal λ, nine features with non-zero coefficients were selected.

**Figure 4 cancers-14-03515-f004:**
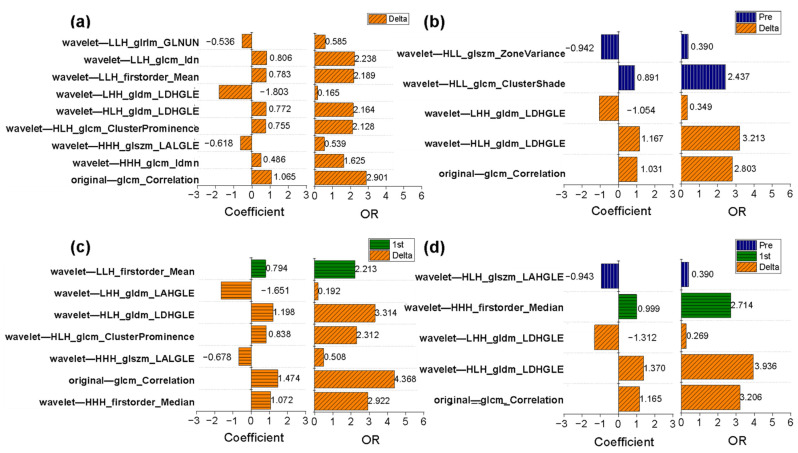
Odds ratio (OR) values and coefficients of the selected radiomic features in the delta-radiomics model and the longitudinal fusion radiomic model based on CEe. (**a**) A total of 9 key features are in the delta-radiomics model, which is the optimal separate radiomic model. (**b**) The selected 5 features were 2 baseline and 3 delta-radiomics features in the fusion radiomic model of pre-radiomics features and delta-radiomics features. (**c**) The fusion radiomic model of 1st-radiomics features and delta-radiomics features retains 7 features, 6 from delta-radiomics features and 1 from 1st-radiomics features. (**d**) A total of 5 features of the fusion radiomic-models of all features based on CEe, and 1 from pre-radiomics features, 1 from 1st- radiomics features, and 3 from delta-radiomics features.

**Figure 5 cancers-14-03515-f005:**
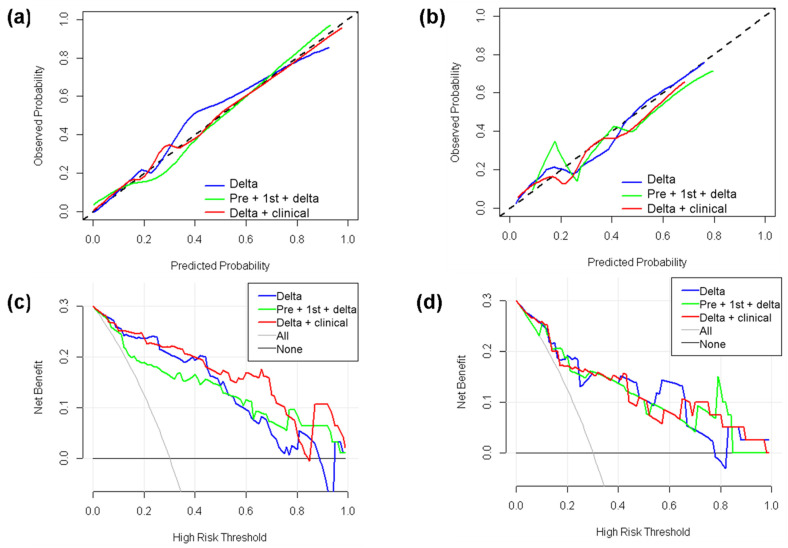
Calibration curves and decision curve analysis (DCA) for the optimal delta-radiomics model, longitudinal fusion-radiomics model, and clinical-radiomics model. (**a**) Calibration curves for the training cohort. (**b**) Calibration curves for the validation cohort. The dotted line represents a perfect classification. (**c**) DCA results for the training cohort. (**d**) DCA results for the validation cohort. The *y*-axis represents the net benefit. The solid gray line represents the scenario that all patients were included in the pCR group. The solid black line represents the scenario that no patients were included in the pCR group. The *x*-axis represents the threshold probability (i.e., where the expected benefit of the treatment is equal to the expected benefit of avoiding treatment).

**Figure 6 cancers-14-03515-f006:**
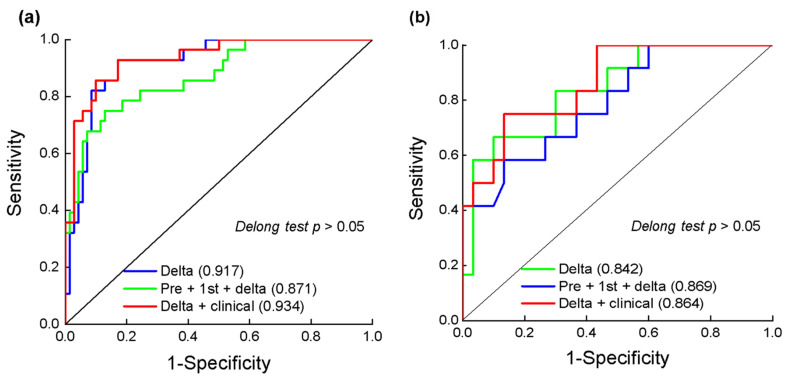
Receiver operating characteristic (ROC) curves of the delta-radiomics model, the longitudinal fusion radiomic model, the clinical-radiomics model in training cohort (**a**) and validation cohort (**b**) based on CEe. Numbers in parentheses are areas under the receiver operating characteristic curve (AUC).

**Figure 7 cancers-14-03515-f007:**
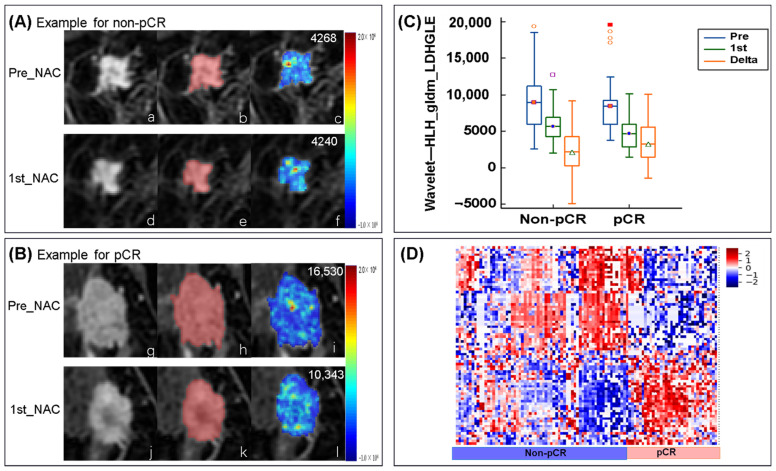
Examples feature of wavelet—HHL_ gldm_ Large Dependence High Gray Level Emphasis (LDHGLE) in non-pCR and pCR. (**A**) Images from a non-pCR breast cancer patient (aged 57 years old with invasive ductal carcinoma of the triple negative subtype. The first line is: (**a**) the image of pre-NAC, (**b**) ROI, and (**c**) a LDHGLE map of tumor ROI (mean LDHGLE value = 4268). The second line is: (**d**) the image of the first cycle of NAC(1st-NAC), (**e**) ROI, and (**f**) a LDHGLE map of tumor ROI (mean LDHGLE value = 4240). (**B**) Images from a pCR breast cancer patient (aged 51 years old with invasive ductal carcinoma of luminal B subtype). The first line is: (**g**) the image of pre-NAC, (**h**) ROI, and (**i**) a LDHGLE map of tumor ROI (mean LDHGLE value = 16,530). The second line is: (**j**) the image of 1stNAC, (**k**) ROI, and (**l**) a LDHGLE map of tumor ROI (mean LDHGLE value = 10,343). (**C**) Boxplot represents the feature of wavelet-HHL_ gldm_ LDHGLE distribution among patients of non-pCR and pCR in the training cohort. (**D**) The heatmap of selected feature in delta-radiomics model based on early phase. Demonstrates overall distribution of key delta-radiomics features among patients with pCR and non-pCR in the training cohort, which shows the obvious difference between the two groups. LDHGLE, large dependence high gray level emphasis.

**Table 1 cancers-14-03515-t001:** Comparison of clinicopathological characteristics in the training and validation cohort.

Characteristics	Training Cohort (*n* = 98)		*p*	Validation Cohort (*n* = 42)		
	pCR (*n* = 28)	Non-pCR (*n* = 70)		pCR (*n* = 12)	Non-pCR (*n* = 30)	*p*
Age (years)	51.0 ± 9.1	51.7 ± 10.5	0.601	50.7 ± 7.2	47.3 ± 10.7	0.242
Tumor size (cm)	4.2 (3.1, 5.6)	4.8 (3.4, 6.1)	0.276	3.4 (2.5, 5.2)	5.0 (3.1, 5.7)	0.177
Enhancement type (%)			0.442			0.657
Mass-like	15 (53.6)	42 (60)		7 (58.3)	14 (46.7)	
Non-mass-like	6 (21.4)	8 (11.4)		2 (16.7)	4 (13.3)	
Mass + non-mass	7 (25.0)	20 (28.6)		3 (25.0)	12 (40.0)	
Type (%)			0.236			0.469
Multi-lesion	8 (28.6)	29 (41.4)		3 (25.0)	11 (36.7)	
Single-lesion	20 (71.4)	41 (58.6)		9 (75.0)	19 (63.3)	
TNM (%)			0.276			0.368
Ⅱ A	7 (25.0)	9 (12.9)		3 (25.0)	7 (23.3)	
Ⅱ B	5 (17.9)	18 (25.7)		5 (41.7)	5 (16.7)	
Ⅲ A	2 (7.1)	11 (15.7)		2 (16.7)	5 (16.7)	
Ⅲ B	9 (32.1)	26 (37.1)		2 (16.7)	10 (33.3)	
Ⅲ C	5 (17.9)	6 (8.6)		0 (0.0)	3 (10.0)	
Grades (%)			0.104			0.205
2	13 (46.4)	45 (64.3)		10 (83.3)	19 (63.3)	
3	15 (53.6)	25 (35.7)		2 (16.7)	11 (36.7)	
ER status (%)			0.001 *			0.002 *
Positive	5 (17.9)	46 (65.7)		2 (16.7)	21 (70.0)	
Negative	23 (82.1)	24 (34.3)		10 (83.3)	9 (30.0)	
PR status (%)			0.001 *			0.554
Positive	7 (25.0)	49 (70.0)		7 (58.3)	17 (56.7)	
Negative	21 (75.0)	21 (30.0)		5 (41.7)	13 (43.3)	
HER2 status (%)			0.041 *			0.008 *
Positive	15 (53.6)	22 (31.4)		8 (66.7)	7 (23.3)	
Negative	13 (46.4)	48 (68.6)		4 (33.3)	23 (76.7)	
Ki-67 status (%)			0.092			0.263
≤20%	2 (7.1)	15 (21.4)		1 (8.3)	23 (76.7)	
>20%	26 (92.9)	55 (78.6)		11 (91.7)	7 (23.3)	
Molecular subtypes (%)			<0.001 *			0.046 *
Luminal A	0 (0.0)	5 (7.1)		0 (0.0)	2 (6.7)	
Luminal B	7 (25.0)	48 (68.6)		5 (41.7)	19 (63.3)	
HER2 enriched	11 (39.3)	5 (7.1)		5 (41.7)	2 (6.7)	
TN	10 (35.7)	12 (17.2)		2 (16.7)	7 (23.3)	

Age is presented as mean ± SD. Tumor size is presented as median (interquartile range), and the others are shown as proportions (percentages). * *p* < 0.05. pCR, pathologic complete response; ER, estrogen receptor; PR, progesterone receptor; HER2, human epidermal growth factor receptor 2; TN, triple negative.

**Table 2 cancers-14-03515-t002:** Diagnostic performance of separate models.

	CEe	CEp	CEd
Pre-radiomics model			
No. of selected features			
LASSO_ CV	9	9	2
Logistic_ CV	3	5	1
AUC (training/validation)	0.759/0.617	0.827/0.694	0.649/0.539
95% CI of AUC	0.647, 0.871/0.403, 0.830	0.723, 0.922/0.533, 0.856	0.527, 0.770/0.319, 0.759
Sensitivity (training/validation)	0.643/0.667	0.679/0.917	0.857/0.250
Specificity (training/validation)	0.800/0.633	0.857/0.500	0.386/0.967
Accuracy (training/validation)	0.755/0.643	0.806/0.619	0.520/0.762
1st-radiomics model			
No. of selected features			
LASSO_ CV	10	9	13
Logistic_ CV	5	4	4
AUC (training/validation)	0.803/0.775	0.816/0.650	0.826/0.703
95% CI of AUC	0.694, 0.913/0.627, 0.923	0.717, 0.915/0.432, 0.868	0.738, 0.914/0.514, 0.892
Sensitivity (training/validation)	0.756/0.667	0.786/0.667	0.821/0.417
Specificity(training/validation)	0.771/0.800	0.771/0.800	0.700/0.967
Accuracy (training/validation)	0.776/0.762	0.776/0.667	0.735/0.810
Delta-radiomics model			
No. of selected features			
LASSO_ CV	13	11	3
Logistic_ CV	9	7	1
AUC (training/validation)	0.917/0.842	0.803/0.764	0.708/0.697
95% CI of AUC	0.861, 0.974/0.709, 0.974	0.64, 0.913/0.592, 0.936	0.594, 0.821/0.512, 0.883
Sensitivity (training/validation)	0.929/0.667	0.786/0.917	0.750/0.833
Specificity (training/validation)	0.829/0.900	0.771/0.667	0.629/0.700
Accuracy (training/validation)	0.857/0.833	0.776/0.738	0.663/0.738

Pre-radiomics features, the features from DCE-MRI before neoadjuvant chemotherapy (NAC); 1st-radiomics features, the features from DCE-MRI after the first cycle of NAC. CV, cross validation. AUC, area under receiver operating characteristic curve; CI, confidence intervals.

**Table 3 cancers-14-03515-t003:** Predictive performance of the longitudinal fusion radiomic models and clinical-radiomics models based on the early phase.

	AUC	Sensitivity	Specificity	Accuracy
Longitudinal fusion radiomic-models (training/validation)				
Pre + delta	0.866/0.750	0.929/0.833	0.623/0.700	0.723/0.738
1st + delta	0.903/0.839	0.821/0.667	0.871/0.933	0.857/0.857
Pre + 1st + delta	0.871/0.869	0.750/0.667	0.871/0.900	0.837/0.833
Clinical-radiomic model (training/validation)				
Pre + clinical	0.762/0.569	0.679/0.333	0.757/0.900	0.735/0.738
1st + clinical	0.801/0.767	0.786/0.833	0.771/0.633	0.776/0.691
Delta + clinical	0.934/0.864	0.927/0.750	0.829/0.867	0.857/0.833
Pre + delta + clinical	0.870/0.736	0.821/0.667	0.757/0.800	0.776/0.762
1st + delta + clinical	0.908/0.828	0.821/0.667	0.900/0.967	0.878/0.881
Pre + 1st + delta + clinical	0.866/0.864	0.714/0.667	0.886/0.900	0.837/0.833

**Table 4 cancers-14-03515-t004:** The nine key radiomic features in the delta-radiomics model and their pre-radiomics and 1st-radiomics features.

Median (IQR)	pCR			Non-pCR			Trend
	Pre	1st	Delta	Pre	1st	Delta	
First order-related features							
LLH_Mean (×10^1^)	−7.2 (−9.5, 4.5)	−7.2 (−14.8, −5.2)	2.6 (−0.5, 5.0)	−6.7 (−9.0, −5.0)	−7.7 (−10.6, 5.5)	0.8 (−1.0, 2.6)	Down
GLCM-related features							
Correlation (×10^−1^)	7.2 (6.2, 8.0)	6.2 (5.2, 7.5)	0.9 (0.03, 1.7)	7.4 (6.5, 8.0)	7.0 (6.0, 7.8)	0.3 (−0.3, 0.8)	Down
HHH_Idmn (×10^−1^)	9.9 (9.9, 10.0)	9.8 (9.8, 9.9)	0.1 (0.03, 0.2)	9.9 (9.9, 10.0)	9.9 (9.8, 10.0)	0.06 (−0.09, 0.08)	Down
HLH_ Cluster Prominence (×10^4^)	3.0 (0.9, 6.2)	1.73 (0.4, 3.0)	1.1 (0.1, 3.4)	2.3 (1.1, 4.2)	2.0 (0.8, 4.2)	0.2 (−1.2, 1.4)	Down
LLH _ Idn (×10^−1^)	9.2 (9.1, 9.3)	9.1 (8.9, 9.2)	0.1 (0.04, 0.2)	9.2 (9.1, 9.3)	9.2 (9.1, 9.3)	0.05 (−0.02, 0.13)	Down
GLDM-related features							
HLH_ LDHGLE (×10^3^)	8.5 (6.0, 12)	4.7 (2.8, 7.6)	3.5 (0.9, 6.0)	9.5 (5.9, 16.0)	6.6 (4.2, 11.2)	2.4 (−0.6, 4.9)	Down
LHH_ LDHGLE (×10^4^)	1.0 (0.6, 1.6)	0.6 (0.3, 1.0)	0.3 (0.003, −1.1)	1.2 (0.7, 1.8)	0.7 (0.6, 2.7)	0.3 (0.03, −0.7)	Down
GLSZM-related features							
HHH_ LALGLE (×10^3^)	0.9 (0.2, 6.3)	0.4 (0.05, 6.1)	0.2 (0.04, 1.1)	1.6 (0.4, 5.9)	0.9 (0.2, 5.0)	0.1 (−0.9, 1.3)	Down
GLRLM-related features							
LLH_ GLNUN (×10^−2^)	0.9 (0.7, 1.2)	1.0 (0.8, 1.3)	−0.1 (−0.3, 0.02)	0.9 (0.7, 1.1)	0.9 (−0.8, 1.2)	−0.03 (−0.2, 0.1)	Up

HHH, HLH, LHH, and LLH indicate that the feature type is a wavelet transform feature. IQR, interquartile range. LALGLE, large area low gray level emphasis; LDHGLE, large dependence high gray level emphasis; GLNUN, gray level non-uniformity normalized.

## Data Availability

The data presented in this study are available on reasonable request from the corresponding author L.Z., zhanglnda@163.com. Due to privacy restrictions the data are not publicly available.

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
