# Peer review of "Delta-Radiomics Based on Dynamic Contrast-Enhanced MRI Predicts Pathologic Complete Response in Breast Cancer Patients Treated with Neoadjuvant Chemotherapy"

_cancers, 2022, doi:10.3390/cancers14143515_

Round 1
Reviewer 1 Report
The authors in this manuscript aim to investigate the value of delta-radiomics after the first cycle of neoadjuvant chemotherapy (NAC) using dynamic contrast-enhanced magnetic resonance imaging (DCE) for early prediction of pathological complete response (PCR) in breast cancer patients.
The paper is well written, the introduction clearly outlines the problem and objectives, the method is well explained, and the results are consistent and interesting.
However, the discussion and references need in my opinion to be further strengthened by comparison with other working methods in NAC response prediction such as deep-learning and transfer-learning. In this regard you can mention these works in the discussion by adding them in your bibliography:
1) Comes MC et al. Early prediction of neoadjuvant chemotherapy response by exploiting a transfer learning approach on breast DCE-MRIs. Scientific Reports, 2021, 11(1), 14123
2) Massafra R et al. Robustness Evaluation of a Deep Learning Model on Sagittal and Axial Breast DCE-MRIs to Predict Pathological Complete
Response to Neoadjuvant Chemotherapy. Journal Pers. Med. 2022, 12, 953. https://doi.org/10.3390/jpm12060953
In addition, another method to predict response to neoadjuvant therapy is BPE (background parenchymal enhancement) with qualitative or quantitative assessment, which there is no mention of in your article. You can include this discussion by referring to this article:
La Forgia D. et al. Response predictivity to neoadjuvant therapies in breast cancer: A qualitative analysis of background parenchymal enhancement in dce-mri. Journal of Personalized Medicine, 2021, 11(4), 256
Author Response
Thanks for the reviewer’s comments. We carefully read the literature recommended by the reviewer. We felt that the difficulties of current research, such as time-consuming manual labeling, inconsistent DCE-MRI protocols, etc., can be overcome by deep learning and transfer learning. They are promising research methods, and also the future direction of our team. In the revised manuscript, we enriched the Discussion section by adding perspectives on deep learning and transfer learning as well as background parenchymal enhancement. The content is: “Recent studies [38,39], which used deep learning and transfer learning for feature extraction and selection without human intervention, have been already successfully applied on pre-treatment and early-treatment DCE-MRI and achieved a good performance. Some difficulties have been recently overcome thanks to deep learning, such as time-consuming manual labeling, inconsistent DCE-MRI protocols, etc. Furthermore, fully automatic segmentation is not restricted to intratumoral features, breast tissue [40] and peritumoral [41] can also give an early prediction. This is the direction of our further efforts.” We hope that such modifications will satisfy the reviewer.
Reviewer 2 Report
This manuscript entitled, “Delta-Radiomics based on DCE-MRI predicts pathologic complete response in breast cancer patients treated with neoadjuvant chemotherapy” evaluates early predictive biomarkers of response following the first cycle of NAC based on post contrast early, peak and delta phases in 140 patients that was split between training (n=98) and validation (n=42). The delta-radiomics model based on earl-phases demonstrated a high AUC of 0.842 which outperformed the pre-radiomics model. Overall, this is a well-described comprehensive manuscript describing the potential to use radiomic features extracted from DCE-MRI to predict response. Well written with excellent figures. Some major and some minor points below to address:
- It is unclear why patients that had a change in chemotherapy regimens were removed from analysis unless it was solely based on toxicity and not progression. If progression of disease, this could be beneficial knowledge
- More information on the temporal resolution of the DCE-MRI scans is required to understand if this could be easily incorporated into clinical workflow or if multisite data could be combined to provide more power. Please also comment on this important aspect in the discussion
- Segmentation accuracy should be the intersect of the two radiologists (not necessarily the more senior). In order to be more conservative, any cancerous tissue should be considered.
- Figure 2 workflow is fantastic
- Needs to be compared to traditional analysis with DCE-MRI that is less complex (SER, Kety-tofts model if temporal resolution is high enough)
- Should also report what the specificity is if you set the sensitivity to 95%
Author Response
Point 1: It is unclear why patients that had a change in chemotherapy regimens were removed from analysis unless it was solely based on toxicity and not progression. If progression of disease, this could be beneficial knowledge.
Response 1: Thanks for the reviewer’s comments. In our study, the earliest treatment time-point – after the first cycle of NAC was used to evaluate the final efficacy of NAC. However, all changes in chemotherapy regimens occurred after the second NAC cycle or even later. This means that the changes of radiomic features after the first cycle of NAC can only reflect the effect of the initial NAC regimen, but cannot represent the final changes and pathological reactions of the subsequent regimen. Therefore, we had to exclude samples in which the chemotherapy regimen had changed.
Point 2: More information on the temporal resolution of the DCE-MRI scans is required to understand if this could be easily incorporated into clinical workflow or if multisite data could be combined to provide more power. Please also comment on this important aspect in the discussion.
Response 2: Thanks to the reviewers for the professional comments. DISCO protocol used in our study has a temporal resolution of 19.4 s. The protocol provides higher temporal and comparable spatial resolution compared with clinical standard protocol, which is beneficial for detection and classification of breast lesions with high accuracy. However, the clinical application of DISCO is still in its infancy and remains exploratory. The first issue to address is the storage and transmission difficulties caused by big data volumes. The absence of clear guidelines for quantitative measurement is another current issue. Related content has been complemented in the Discussion Section: “Despite the noted strengths, the clinical application of DISCO is still in its infancy and remains exploratory. The first issue to address is the storage and transmission difficulties caused by big data volumes. The absence of clear guidelines for quantitative measurement is another current issue.”
We are sorry that we don’t understand what "multisite" means. If it means a combination of multiple contrast phases of DCE-MRI, we think it is possible to further improve the diagnostic performance in quantitative or radiomics studies. We will conduct further verification in the future. Thank you for pointing the way to our future research.
Point 3: Segmentation accuracy should be the intersect of the two radiologists (not necessarily the more senior). In order to be more conservative, any cancerous tissue should be considered.
Response 3: Thank you for your comments. We are sorry that the description in previous manuscript was inappropriate. Before all VOI delineation, we performed an ICC test, and found that the two radiologists had good consistency, so we chose the same radiologist for all the VOI delineations. We have made changes in the revised manuscript: “An ICC value of 0.8 or greater was considered to indicate almost perfect consistency, and the feature was retained. Features with ICC values less than 0.8 were initially eliminated. Then, the VOIs delineated by the radiologist with two years of experience were used as the final segmentation.”
Radiomics can extract quantitative features using advanced feature analysis with high throughput from digital medical images. If all the lesions are delineated in multifocal and multicentric breast cancer, the radiomic features will be more skewed distributions, such as NGTDM-related features, which quantify the difference between a gray value and the average gray value of its neighbours within distance δ. This will affect the accuracy of the final data. In the following articles, the largest lesion was also selected in multifocal and multicentric breast cancer.
- Kuramoto, Y.; Wada, N.; Uchiyama, Y. Prediction of pathological complete response using radiomics on MRI in patients with breast cancer undergoing neoadjuvant pharmacotherapy. Int J Comput Assist Radiol Surg 2022, doi:10.1007/s11548-022-02560-z.
- Li, Q.; Xiao, Q.; Li, J.; Wang, Z.; Wang, H.; Gu, Y. Value of Machine Learning with Multiphases CE-MRI Radiomics for Early Prediction of Pathological Complete Response to Neoadjuvant Therapy in HER2-Positive Invasive Breast Cancer. Cancer Management and Research 2021, Volume 13, 5053-5062, doi:10.2147/cmar.S304547.
Point 4: Figure 2 workflow is fantastic
Response 4: Thank you for your comments. It is our honor to receive your compliment.
Point 5: Needs to be compared to traditional analysis with DCE-MRI that is less complex (SER, Kety-tofts model if temporal resolution is high enough).
Response 5: We strongly agree with the reviewer. We only extracted radiomic features without further analysis of model-based quantitative parameters. This is one of the limitations of our study. We added it in the Limitation Section: “Fourth, DISCO protocol with high spatiotemporal resolution is conducive to quantitative and semi-quantitative measurements, while we did not compare or combine radiomics with quantitative parameters.” In the future, we will accumulate more samples and continue this study. Thank you again for pointing the way for our study.
Point 6: Should also report what the specificity is if you set the sensitivity to 95%.
Response 6: Thank you for your comments. I’m sorry for the trouble caused by the unclear description in the previous manuscript. We have made changes to Table 2. In Table 2, the 95% CIs are for AUC values rather than sensitivity.
Reviewer 3 Report
Dear Authors,
I congratulate you on the interesting topic. You provided a well-written prospective study to evaluate the usefulness of delta-radiomics for early prediction of pathological complete response (pCR) in patients with breast cancer. You found statistically significant results confirming the hypothesis that the delta-radiomics features reflected the therapy-induced changes in tumor heterogeneity, allowing the prediction of pCR. Furthermore, you confirmed the importance of the early phase of DCE-MRI as the phase in which the response to NAC could be better evaluated, subsequently proposing the more relevant features to be considered.
Even with some limitations, the paper is very interesting, original and it could be considered as a first relevant step towards an in-depth evaluation about the use of these techniques.
I suggest to better express some concepts of the introduction section. Indeed, some sentences are unclear of shortened without explanations. For example, lines 87-88 refers to your purpose or are they representative of the results of the study?
Given these considerations, after a minor spell check, the contribution of the paper is sufficiently developed such that it could be accepted for publication.
Kind Regards
Author Response
Point 1: I suggest to better express some concepts of the introduction section. Indeed, some sentences are unclear of shortened without explanations. For example, lines 87-88 refers to your purpose or are they representative of the results of the study?
Response 1: Thank you for your comments. We have carefully checked the manuscript and adjusted some unclear descriptions. We adjusted lines 87-88 to “Additionally, we also used differential subsampling with cartesian ordering (DISCO) DCE-MRI, which has high spatiotemporal resolution [26], to analyze the model performance in the early, peak, and delay phases during contrast agent inflow and outflow, and determined the optimal contrast phase of DCE-MRI.”
This manuscript was edited in the English language and reviewed by AiMi Academic Services (www.aimieditor.com), and the language assistant certificate is in the attachment.
